# Nitrogen Acquisition and Transport in the Ectomycorrhizal Symbiosis—Insights from the Interaction between an Oak Tree and *Pisolithus tinctorius*

**DOI:** 10.3390/plants12010010

**Published:** 2022-12-20

**Authors:** Mónica Sebastiana, Susana Serrazina, Filipa Monteiro, Daniel Wipf, Jérome Fromentin, Rita Teixeira, Rui Malhó, Pierre-Emmanuel Courty

**Affiliations:** 1BioISI—Instituto de Biosistemas e Ciências Integrativas, Faculdade de Ciências, Universidade de Lisboa, 1749-016 Lisboa, Portugal; 2Linking Landscape, Environment, Agriculture and Food (LEAF), Associated Laboratory TERRA, Instituto Superior de Agronomia (ISA), Universidade de Lisboa, 1349-017 Lisbon, Portugal; 3Centre for Ecology, Evolution and Environmental Changes (cE3c) & CHANGE—Global Change and Sustainability Institute, Faculdade de Ciências da Universidade de Lisboa, Campo Grande, 1749-016 Lisboa, Portugal; 4Agroécologie, INRAE, Institut Agro, University Bourgogne Franche-Comté, F-21000 Dijon, France

**Keywords:** mycorrhiza, nitrogen, ammonium transporter, ectomycorrhizal fungi, oak (cork oak)

## Abstract

In temperate forests, the roots of various tree species are colonized by ectomycorrhizal fungi, which have a key role in the nitrogen nutrition of their hosts. However, not much is known about the molecular mechanisms related to nitrogen metabolism in ectomycorrhizal plants. This study aimed to evaluate the nitrogen metabolic response of oak plants when inoculated with the ectomycorrhizal fungus *Pisolithus tinctorius*. The expression of candidate genes encoding proteins involved in nitrogen uptake and assimilation was investigated in ectomycorrhizal roots. We found that three oak ammonium transporters were over-expressed in root tissues after inoculation, while the expression of amino acid transporters was not modified, suggesting that inorganic nitrogen is the main form of nitrogen transferred by the symbiotic fungus into the roots of the host plant. Analysis by heterologous complementation of a yeast mutant defective in ammonium uptake and GFP subcellular protein localization clearly confirmed that two of these genes encode functional ammonium transporters. Structural similarities between the proteins encoded by these ectomycorrhizal upregulated ammonium transporters, and a well-characterized ammonium transporter from *E. coli*, suggest a similar transport mechanism, involving deprotonation of NH_4_^+^, followed by diffusion of uncharged NH_3_ into the cytosol. This view is supported by the lack of induction of NH_4_^+^ detoxifying mechanisms, such as the GS/GOGAT pathway, in the oak mycorrhizal roots.

## 1. Introduction

Mycorrhizal symbiosis is known to improve the nutrient status of host plants. Hyphae from mycorrhizal fungi can very efficiently explore the rhizosphere, absorbing nutrients which are then transferred to the plant partner in exchange for photosynthetic-derived carbon compounds. Ectomycorrhiza is the dominant plant symbiotic interaction in temperate forest ecosystems, where nitrogen (N) availability is one of the major factors limiting plant growth [1]. This is usually the result of a reduced bacterial nitrification activity in these regions and the sequestration of inorganic N to humic particles [2]. It is estimated that 80–90% of trees in temperate and boreal forests form beneficial symbiotic relationships with ectomycorrhizal (ECM) fungi [3], which drive soil processes such as nutrient cycling, organic matter decomposition, soil aggregation, and carbon sequestration [4]. Several studies have provided evidence that, in these ecosystems, most trees rely on their symbiotic partners to acquire N (e.g., [5]). N is an essential nutrient for plant growth, being part of most organic macromolecules, such as proteins, nucleic acids, hormones, and vitamins. It can occur in the soil either in inorganic form, nitrate (NO_3_^−^) and ammonium (NH_4_^+^), or as organic compounds, such as amino acids. However, the inorganic forms are dominant in the soil. Studies have shown that ECM fungi can metabolize both organic (such as amino acids) and inorganic N forms (such as NH_4_^+^ and NO_3_^−^) [6]. Regarding inorganic N, these fungi seem to preferentially use NH_4_^+^, since NO_3_^−^ needs to be immediately reduced to NH_4_^+^ after uptake, which increases the energetic costs for N assimilation [5]. Several ammonium importers have been identified and characterized in several ECM fungal species [4].

According to current knowledge from arbuscular mycorrhizas, the extraradical mycelium of mycorrhizal fungi can take up inorganic and organic N from the soil, which is then assimilated by the GS/GOGAT pathway into amino acids, such as arginine [7,8]. Arginine is then translocated to the symbiotic intraradical hyphae, where it is converted to NH_4_^+^, which is thought to be the main N form transferred by arbuscular mycorrhizal fungi to their host plants [9,10]. Symbiotic N transfer by ECM fungi is less clear, but current knowledge points to NH_4_^+^ and/or amino acids as the main N forms involved [11,12]. The idea that amino acids could constitute a potential N source delivered by ECM fungi is based on the fact that some ECM fungi can upregulate their amino acid exporters in ECM root tips [13,14], as well as on indirect evidence from ^15^N labeling experiments showing that labelled amino acids are found in ECM plants following the application of labeled glycine to the humus layer [15]. However, fungal N delivery as amino acids would result in a severe carbon loss by fungal hyphae [16] and, since the pH of the apoplastic plant/fungus interface is rather low (3.1–4.0), a reduced amino acid excretion ability of fungal hyphae would be expected [17].

Studies have shown that NH_4_^+^ can be exported by intraradical hyphae during arbuscular mycorrhizal symbiosis [7]. Transfer of NH_4_^+^ from the fungal hyphae into the symbiotic interface is thought to be mediated by fungal plasma membrane transporters, followed by uptake by plant ammonium transport proteins (AMT) localized at the root cell plasma membrane [5]. The family of AMT proteins has been characterized mostly on plants establishing arbuscular mycorrhizas, with several AMTs being highly expressed at the root–fungus interface of mycorrhizas of *Sorghum bicolor* [18], *Oryza sativa* [19], *Glycine max* [20], *Lotus japonicus* [21], and *Medicago truncatula* [22]. Some AMTs have been shown to be specifically over-expressed in root cells containing arbuscules, suggesting a role in the transport of fungal-derived NH_4_^+^ into the root cell cytoplasm [20].

In *Poplar* and *Pinus*, ECM root colonization results in a significant induction of AMT gene expression, suggesting that these transporters are also involved in the ECM incorporation of NH_4_^+^ exported by the fungus into the root cells [17,23]. Due to their ecological importance, there is a need to improve our understanding on the molecular mechanisms driving nutrient exchange dynamics in the ECM roots of forest trees so that we can better predict their role in tree adaptation to climate change scenarios. Overall, while the transport of N from fungal to plant tissues is certainly an accepted process, neither the form of N transported, nor the specific fungal or plant transport mechanisms are fully known [4]. In the present study, the roots of cork oak (*Quercus suber*) plants inoculated with the ECM fungus *Pisolithus tinctorius* were used as a model system to investigate the molecular basis of N uptake and transfer in the ECM plant–fungus system. We focused on several oak genes that may be involved in the transfer of organic and inorganic N forms from the fungus to the host plant. In particular, we took advantage of the cork oak genome information to identify genes encoding AMTs and transporters involved in root amino acid uptake (such as lysine/histidine transporters and amino acid permeases) [24]. The transcript level of these genes was evaluated in oak roots in the presence or absence of the ECM fungus *P. tinctorius* in order to identify and functionally characterize specific AMTs and amino acid transporters regulated by the ECM interaction and define their pattern of expression during symbiosis.

## 2. Results

### 2.1. P. tinctorius Root Colonization. Effect on Plant Growth and N Concentration

Root colonization of cork oak plants, 17 months after *P. tinctorius* inoculation, measured as % of root segments showing mycorrhizal root tips, was of about 40% (Appendix A). N concentration in leaves was significantly increased by the ECM symbiosis, *P. tinctorius* inoculation, resulting in 5.4% additional N in inoculated plants grown with no N fertilization (Appendix A). Despite this, *P. tinctorius*-inoculated young cork oak plants (21 months old) did not show significant improvements in growth, measured as plant height and shoot and root fresh weight, under our experimental conditions (Appendix A).

### 2.2. Identification of Cork Oak Genes Involved in N Transport and Assimilation

The genome sequence of *Q. suber*, available at the NCBI database, was searched for genes potentially involved in the uptake of inorganic (NH_4_^+^) and organic (amino acids) N forms. Twenty genes corresponding to AMTs were identified based on NCBI searches using keyworks and similarity with protein sequences from *Arabidopsis* and *Populus*. Three genes were discarded since the protein sequences had no significant similarity against the typical ammonium transporter family domain (pfam00909) in Pfam. Another set of seven genes was also discarded since its protein sequences showed similarity only to AMTs from ascomycetes. In total, 10 genes encoding *Q. suber* AMTs were selected for further analysis (Appendix A). In addition, five genes were identified with high similarity to membrane proteins described in other plants as involved in root amino acid uptake from soil, including two amino acid permeases, two lysine histidine transporters, and one proline transporter protein (Appendix A). A search against the cork oak genome allowed the identification of genes encoding proteins involved in the GS/GOGAT pathway for NH_4_^+^ assimilation, including two cytosolic isoforms of glutamine synthase (GS1 and GS1.3, respectively), a chloroplastic GS isoform (GS2), a chloroplastic NADH-dependent glutamate synthase (GOGAT) isoform (NADH-GOGAT), and the ferredoxin-dependent amyloplastic (Fd-GOGAT) isoform (Appendix A).

In silico analysis of the protein sequences (Appendix A) shows that all the cork oak AMTs contained the protein domains characteristic of AMT protein family (e.g., IPR001905). Prediction analysis of amino acid sequences indicated they are membrane proteins, with a signal peptide sequence and 11 transmembrane domains (except QsAMT1.1-like (b), which has 9) with an extracellular N-terminus and a cytosolic C-terminus, like other plant AMT members [25,26]. Regarding the putative amino acid transporters, all the cork-oak-selected genes showed the conserved amino acid transporter transmembrane domain, a signal peptide sequence, and 9–11 predicted transmembrane domains, indicating a protein localization at the plasma membrane, consistent with their role in root amino acid uptake from soil. Sequences of the proteins involved in the GS/GOGAT pathway showed the glutamine synthase and glutamate synthase domains, respectively, and a globular conformation with no transmembrane domains, characteristic of cytoplasmic proteins.

### 2.3. Impact of ECM Symbiosis on the Expression of Genes Involved in N Transport and Assimilation

To evaluate the effect of ectomycorrhiza formation on the expression of cork oak AMTs, amino acid importer genes, and genes involved in the GS/GOGAT cycle, specific primer pairs were designed for qPCR analysis to compare *P. tinctorius* inoculated and non-inoculated plants. Root samples from 21-month-old inoculated and non-inoculated cork oak plants, collected at 17 months post-*P. tinctorius* inoculation, were used for analysis. Leaves from each treatment were also analyzed to evaluate tissue specificity of gene expression. The transcript levels of three AMT genes were below the detection limit under our experimental conditions (*QsAMT1.1-like* (b), *QsAMT3.1-like* (b), and *QsAMT3.3-like*).

The remaining seven cork oak AMT genes were expressed in roots, with some also being expressed in leaves but at lower levels, except for *QsAMT1.1-like a*, whose expression was similar to those observed in roots, as well as *QsAMT3.3-like* (a), which was more expressed in leaves of non-inoculated plants (Figure 1).

Ectomycorrhiza formation with *P. tinctorius* significantly increased the expression of three cork oak AMTs in inoculated roots (*QsAMT1.2-like*, *QsAMT3.2-like*, and *QsAMT3.3-like* (a)) compared with roots of the control treatment (Figure 1). However, the expression level of one AMT, *QsAMT1.3-like*, was decreased in roots of ECM cork oak plants when compared with non-inoculated roots (Figure 1). Regarding *P. tinctorius*’ effect on leaves, most AMTs were not affected by the inoculation, except *QsAMT3.3-like* (a), which was downregulated in inoculated vs. non-inoculated plants (Figure 1).

None of the five cork oak amino acid importers investigated were affected by the ECM interaction with *P. tinctorius* at the transcript level (Appendix A). Among the five tested genes, three had a higher expression level in root tissues relative to leaves (*QsAAP3-like*, *QsLHT1-like*, and *QsProT2*), while two were more expressed in leaves (*QsAAP6-like* and *QsLHT1*). Since several AMT genes were upregulated in ECM roots, we decided to investigate if the GS/GOGAT pathway would be involved in the assimilation of NH_4_^+^ provided by the fungus. Most analyzed genes encoding proteins from the GS/GOGAT pathway were not affected by ECM symbiosis, and only one gene was downregulated in ECM roots (Appendix A). In our experimental conditions, it was not possible to detect transcripts of *QsGS1.3*.

### 2.4. Phylogenetic Analysis of Cork Oak AMTs

In the phylogenetic analysis (Figure 2), four out of the ten cork oak AMT proteins were found to be members of the AMT1 subfamily, while the other six were members of the subfamily AMT2, distributed in two separate clusters, AMT2 and AMT3. No AMT4 members were found in the cork oak genome. The ECM upregulated cork oak AMTs are in the same clusters as other previously characterized AMT, also over-expressed in several plants during mycorrhizal symbiosis with ectomycorrhizal and arbuscular mycorrhizal fungi, such as *SbAMT3.1* [18,27], *OsAMT3.1* [27], *PtrAMT 1.2* [10], *GmAMT4.4, GmAMT1.4* and *GmAMT3.1* [20], and *PttAMT1.2* [17]. The upregulated *QsAMT1.2-like* is closely related to the *Populus trichocarpa* and *Populus tremula×tremuloides AMT1.2* genes, which are both over-expressed during ECM symbiosis [10,17].

### 2.5. Heterologous Complementation of a Yeast Mutant Defective in NH_4_^+^ Uptake

A yeast mutant complementation test was used to demonstrate the NH_4_^+^ transport function and to biochemically characterize the cork oak AMTs which were over-expressed in *P. tinctorius* inoculated vs. non-inoculated roots (*QsAMT1.2-like*, *QsAMT3.2-like*, and *QsAMT3.3-like* (a)). All three transporters were expressed through the yeast expression vector pDR196 [28] in a mutant yeast strain, 31019b, which lacks the three endogenous NH_4_^+^ transporter genes (*MEP1*, *MEP2*, and *MEP3*) and is unable to grow on medium containing <3 mM NH_4_^+^ as the sole N source [29]. *QsAMT1.2-like* and *QsAMT3.2-like* were able to efficiently restore the NH_4_^+^ uptake ability of the yeast mutant, promoting growth of the yeast cells under low NH_4_^+^ (1 mM and 2 mM NH_4_^+^) concentration in the growing media (Figure 3), demonstrating that they encode functional AMTs. *QsAMT3.2-like* complemented more efficiently the mutant yeast phenotype, demonstrated by the better growth of colonies at 1 mM and 2 mM NH_4_^+^ as sole nitrogen source (Figure 3). In contrast, *QsAMT3.3-like* (a) was unable to complement the NH_4_^+^ transport deficiency of the 31019b yeast strain, with no cell growth observed under low NH_4_^+^ (Figure 3). This could be related to ammonium toxicity when QsAMT3.3 was constitutively overexpressed in yeast, as described by Hess and co-workers [30].

Functional expression of QsAMT3.2 and QsAMT1.2 in the yeast triple mutant revealed them to be a high-affinity transporter (HATS) with an apparent Km of 0.6 µM (Appendix A) and a low-affinity transporter (LATS) with an apparent Km of 667 µM, respectively (Appendix A). As for the drop test, we were unable to observe a stable methylamine uptake for QsAMT3.3.

### 2.6. Subcellular Localization of Cork Oak AMTs

For the subcellular localization of the AMTs which were upregulated in *P. tinctorius* inoculated root tissues (*QsAMT1.2-like*, *QsAMT3.2-like*, and *QsAMT3.3-like* (a)), we transiently expressed their coding sequences with the reporter gene GFP under the constitutive 35S promoter. We used infiltrated epidermal leaves of *Nicotiana benthamiana*, considering the high expression levels and easiness of procedure. Out of the three genes tested, we were unable to express *QsAMT3.3-like* (a) under our experimental conditions, which could be related to the lack of a cork-oak-specific factor(s) needed for the proper protein maturation and transport into the plasma membrane in the *N. benthamiana* heterologous system. *QsAMT1.2* was detected exclusively at the plasma membrane (Figure 4a–c), occasionally forming patches and suggesting asymmetric ion transport; the 3D maximal intensity projection picture (Figure 4a) further suggests that these channels span the plasma membrane. In contrast, *QsAMT3.2* displays a distinct pattern from *QsAMT1.2*, with well-distinguished and stronger punctate patches (Figure 4d–f) but also a cytoplasmic reticulate-like distribution, suggestive of protein transport through endoplasmic reticulum and Golgi.

### 2.7. Homology Modeling of AMTs Upregulated in Cork Oak ECM Roots

To further characterize the three AMTs from cork oak over-expressed during symbiosis with *P. tinctorius* (*QsAMT1.2-like*, *QsAMT3.2-like*, and *QsAMT3.3-like* (a)), we compared their deduced protein structure with the X-ray structure of the bacterial homolog AmtB from *E. coli*, which has been fully experimentally and computationally characterized [31]. Comparison between the upregulated cork oak AMTs and the *E. coli* AmtB shows an average sequence similarity of about 30% (Appendix A). The *E. coli* AmtB is an ion channel that recruits NH_4_^+^ at the periplasmic NH_4_^+^ binding site, subsequently deprotonating NH_4_^+^ before translocating neutral NH_3_ into the bacterial cell cytoplasm. Most of the amino acids which have a key role in the transport of NH_4_^+^ along the permeation pore of EcAmtB are conserved in the cork oak AMTs which are upregulated by ECM symbiosis with *P. tinctorius* (Table 1 and Appendix A).

## 3. Discussion

It has been well established that ECM fungi play a key role in N nutrition of their host trees that inhabit ecosystems characterized by relatively N poor soils. However, there is little knowledge about the molecular events related to N metabolism in ECM plants. Furthermore, the organic (amino acids)/inorganic (NH_4_^+^) nature of the N molecules transferred by ECM fungi into their host plants is still a topic of controversy.

Analysis of the cork oak genome identified the presence of ten AMTs, four in the AMT1 subfamily and six in the AMT2 subfamily, which is composed of three clades, AMT2, AMT3, and AMT4. Phylogenetic analysis revealed that the three cork oak AMT genes upregulated in ECM roots belong to different AMT subfamilies and are closely related to other AMTs over-expressed in arbuscular mycorrhizal and ECM plants, such as those in the AMT3 cluster from the *Poaceae* plant family [27], or the AMT1 cluster from the woody plant *Populus* [17]. One of the upregulated AMTs in cork oak (*QsAMT1.2-like*) is a member of the AMT1 cluster, whose members are also over-expressed during the interaction of tree species with ECM fungi, such as *Populus* (*PtrAMT1.2*/*PttAMT1.2*) [10,17] and *Pinus pinaster* (*PpAMT1.3*) [23]. The fact that no members of the AMT1 subfamily have been detected in arbuscular mycorrhizal roots might indicate a specific recruitment of the AMT1 subfamily for the transport of NH_4_^+^ delivered by ECM fungi. These results suggest that *QsAMT1.2-like*/*PtrAMT1.2*/*PttAMT1.2/PpAMT1.3* are the orthologues most responsible for NH_4_^+^ uptake in ECM symbiosis. It has been proposed that the AMT1 subfamily is the most ancient one, having been separated from the AMT2 clade early during the evolution of land plants, after their separation from algae [32].

The other upregulated AMTs from cork oak are members of the AMT3 subfamily, like what is observed on several plant species colonized by arbuscular mycorrhizal fungus, suggesting that AMT3 members correspond to AMTs activated in ECM and arbuscular mycorrhizal roots. One notable exception is the lack of gene members of the AMT4 clade in the genome of cork oak, in contrast with other dicots, such as soybean (6) and *Populus* (4), suggesting a diversified organization of the AMT genes in the dicot plant group. Several AMT4 subfamily members are among the AMTs which are over-expressed in mycorrhizas of several plant species, such as sorghum [18], soybean [20], and rice [27]. However, in *Populus*, there are no indications of an over-expression of AMT4 members during mycorrhiza formation [10], and in several species from the *Poaceae* family, loss of the AMT4 gene function had no consequences on symbiotic N uptake, suggesting that AMT4 has acquired new functions [27].

In contrast with AMTs, root amino acid transporters were not upregulated by the cork oak–*P. tinctorius* symbiosis, adding to the growing body of evidence that inorganic NH_4_^+^, and not organic N in the form of amino acids, could be the main N form delivered by ECM fungi to their host plants. In our study, we focused on amino acid transporters known to be active in roots. However, given the high number of members from this gene family in plants, the transport of organic N by other amino acid transporters not analyzed in the present study cannot be discarded. In addition, the amino acid transporters investigated here could be regulated not at the transcription level but at some other different level such as translationally, functionally or by their localization, making it impossible to detect any changes in our experiment.

Delivery of fungal inorganic N into the plant roots is corroborated by the finding that NH_4_^+^ is excreted by fungal hyphae at intraradical arbuscules in the arbuscular mycorrhizal symbiosis [7], AMT proteins being highly expressed at the intraradical mycelium of arbuscular mycorrhizal fungi [33]. Overall, our results suggest a complementary role of these AMTs in the transport of NH_4_^+^ exported by *P. tinctorius* into the cork oak roots, like it has been proposed in other mycorrhizal plants, where higher expression of several AMT genes has been detected in the roots of plants colonized by arbuscular mycorrhizal fungi [19,20,27] and ECM fungi [17,23]. In accordance with other studies [10,18,27], most cork oak AMTs were found to be root-specific, their levels in leaves being very low or even undetectable. However, one AMT, *QsAMT3.3 a*, had high expression levels in the leaves of non-inoculated plants. This indicates that this gene is mostly active in the shoot (leaves), like some AMTs from other plants which are involved in the acquisition of ammonium from roots [34]. The decreased expression of this AMT in the leaves of *P. tinctorius* inoculated plants constitutes an indication that ECM fungi can systemically regulate the expression of the N transport pathway. A lower expression could be indicative of a reduction in NH_4_^+^ unloading from the xylem at the leaves of *P. tinctorius*-inoculated plants. However, additional work would be necessary to draw any further conclusions.

To confirm the function of the three upregulated AMTs as functional NH_4_^+^ transporters, we analyzed their ability to complement a yeast strain defective in NH_4_^+^ transport. The yeast mutant strain was transformed with the coding sequence of each AMT, which was sufficient to confer the ability to grow under low NH_4_^+^ concentration to two of the cork oak AMTs, confirming that those are functionally involved in NH_4_^+^ uptake. Among these, *QsAMT3.2-like* showed a very efficient ability to complement the lack of NH_4_^+^ transport of the mutant yeast strain, like other AMT3 gene members from plant species, such as sorghum and maize colonized by arbuscular mycorrhizal fungi [27], suggesting a high activity of this AMT in the transport of NH_4_^+^ from mycorrhizal symbiotic fungi. Furthermore, the GFP subcellular localization of two mycorrhiza-activated AMTs in *N. benthamiana* epidermal cells confirmed a plasma membrane localization, in agreement with their putative function as root ion transporters.

Comparison with *AmtB* from *E. coli* (*EcAmtB*) enabled us to obtain some insights about the transport mechanism of the cork oak AMT proteins which are upregulated by the ECM interaction with *P. tinctorius*. According to Wang et al. [35], the EcAmtB protein is a membrane pore that deprotonates NH_4_^+^ at the periplasmic binding side, before translocating it as neutral NH_3_ to the cytoplasmic side. Our analysis shows that key residues are conserved between EcAmtB and the cork oak AMTs. These include the amino acids recruiting NH_4_^+^ from the periplasmic space, those involved in receiving the H^+^ resulting from NH_4_^+^ deprotonation, and those involved in the stabilization of NH_3_ molecules through hydrogen bonding during transport in the permeation pore [31,35]. Some amino acid replacements relative to EcAmtB were detected, such as the residue Ser219, which forms a hydrogen bond with NH_4_^+^ and, in QsAMT3.2-like and QsAMT3.3-like (a), is replaced by an aspartic acid (Asp). This replacement of a neutral serine to a negatively charged aspartic acid was also detected in several plant species establishing arbuscular mycorrhizas and could result in an increased NH_4_^+^ binding affinity [27].

These results support a similar transport mechanism between the *E. coli* EcAmtB and the cork oak AMTs suggested by our study and others [21,27], in which the NH_4_^+^ delivered by the fungus into the plant–fungus apoplastic interface would be deprotonated, being then transported as neutral NH_3_ into the root cells cytoplasm. The uptake of the uncharged molecule NH_3_ would prevent a toxic NH_4_^+^ accumulation in the cytosol of root cells. This agrees with our results, since no activation of NH_4_^+^ detoxifying mechanisms, such as the GS/GOGAT assimilatory pathway which incorporates NH_4_^+^ into amino acids, was observed in the *P. tinctorius* ECM cork oak roots under our conditions. The protons resulting from NH_4_^+^ deprotonation would remain in the plant–fungus interface, maintaining or even reinforcing the acidic pH at that location, providing energy for H^+^-dependent transport processes [21]. This mechanism is consistent with a role of NH_4_^+^ as a major source of ECM fungus-derived N, since a reduced amino acid excretion ability by fungal hyphae is expected to occur at the low pH values of the apoplastic space at the plant–fungus interface [17]. This is further corroborated by the findings that, in soybean and sorghum, some arbuscular mycorrhizal-induced AMTs are specifically expressed in root cells containing arbuscules, where nutrients are released to the plant–fungus interface before being taken up by the plant [18,20]. Studies on the arbuscular mycorrhizal fungus *Rhizophagus irregularis* have shown that fungal AMT genes are highly expressed in the intraradical mycelium of fully established symbiotic roots, pointing to a role of fungal AMTs as export carriers for NH_4_^+^ from the arbuscules to the plant–fungus interface space [33,36]. However, since current evidence suggests that AMTs are involved in the uptake of NH_4_^+^**,** and that NH_4_^+^ seems to be the N form taken up by the plant at the symbiotic interface, expression of fungal AMTs in the arbuscules indicates there might exist a competition between plant and fungus for the N available at the plant–fungus interface [33].

In summary, our results indicate that NH_4_^+^ is probably the main N form exported by the ECM fungus *P. tinctorius* and taken up by the plant root cells during symbiosis. Moreover, no experimental evidence was found for a role of amino acids in the transfer and uptake of N in the ECM symbiotic roots. Our results on AMTs constitute a validation of knowledge already acquired from a few ECM models studied so far [17,23], extending data on root amino acid transport gene expression which, to our knowledge, was not previously investigated in ectomycorrhizas. However, it must be noted that our strategy for candidate gene selection might have missed other unexpected genes or genes not regulated at the transcription level, such as those regulated post-translationally, which could also be involved in the transport of N at the plant–fungus interface. Three AMTs were identified in the cork oak roots which are potentially involved in the transfer of fungal-derived N. Future studies should focus on the regulation of these cork oak AMTs, as well as on gene expression profiling during the interaction with different ECM fungal species and under more natural, e.g., field conditions.

## 4. Materials and Methods

### 4.1. ECM Symbiosis Establishment and Plant Growth

*P. tinctorius* (strain Pt23) inoculum was produced in vitro (from monoxenic cultures established in BAF medium containing 1% glucose) in a mixture of vermiculite and peat irrigated with the same medium, under dark conditions, at 23 °C [37]. *Q. suber* seed germination, plant growth, and *P. tinctorius* inoculation were performed as described earlier [37]. Inoculated and non-inoculated plants (controls) were grown in a greenhouse under a randomized block design, in 10 L pots with a two-week irrigation regime, and with no fertilization applied [37]. Plants with fully established ECM roots (17 months after *P. tinctorius* inoculation) were collected for analysis. Shoots and roots from 9 inoculated and non-inoculated plants were measured and weighted for plant growth determination (height and root/shoot fresh weight). The roots from 3 inoculated and non-inoculated plants were cut into segments (1 cm) and observed under a stereoscopic microscope for determining the average root colonization, calculated as the number of segments showing mycorrhizas relative to the total number of segments analyzed (100). Significant differences in plant biomass and root colonization among treatments were tested by Student’s *t*-test, using the IBM SPSS statistics 26 (Chicago, IL, USA). Lateral roots containing mycorrhizas, corresponding roots from non-inoculated plants, and leaves from inoculated and non-inoculated plants were snap frozen in liquid nitrogen and stored at −80 °C.

### 4.2. Determination of N Concentration

Frozen leaf material from six different ECM and non-ECM plants was dried at 70 °C for 72 h and grounded in a mill (Retsch, Haan, Germany) to a homogenous fine powder for isotopic analysis. After grinding, samples were used for N percentage calculation, according to [38], on a EuroEA 3000 Elemental Analyzer (EuroVector, Milan, Italy) with a TDC detector at the Stable Isotopes and Instrumental Analysis Facility, Faculty of Sciences, Lisbon University. N concentration was defined as % of dry weight. Significant differences among treatments were tested by Student’s t-test, using the IBM SPSS statistics 26 (Chicago, IL, USA).

### 4.3. Identification of Cork Oak Genes Involved in N Transport and Assimilation

Sequencing, assemblage, and annotation of the *Q. suber* genome were described by [39], and all sequences are available at public databases, such as the NCBI. Putative *Q. suber* genes involved in root amino acid transport, such as amino acid permeases (AAP) and lysine/histidine transporters (LHTs), along with genes involved in NH_4_^+^ transport (AMTs) and N assimilation (GS/GOGAT pathway) [24], were identified by searching the NCBI database (restricted to *Q. suber*) using keywords and protein sequences (blastp) from *Arabidopsis* and *Populus* downloaded from TAIR [40] and Phytozome [41], respectively; sequences with E-value < 10^−6^ and bit score > 50 were retained for further analysis. Additional confirmation of the putative *Q. suber* homologs that were detected was performed by identification of the characteristic protein domains, such as the transmembrane amino acid transporter (pfam01490) and the ammonium transporter family domain (pfam00909) in Pfam [42], with protein sequences showing an E-value < 10^−50^ being retained for further analysis. The selected cork oak genes were characterized based on the presence of signal peptides and transmembrane helices with SignalP 6.0 [43] and DeepTMHMM [44]. The subcellular location was predicted with TargetP 2.0 [45].

### 4.4. Phylogenetic Analysis of Q. suber AMT Gene Family

The amino acid sequences of the AMT family members from poplar, rice, sorghum, soybean, tomato, *Lotus japonicus*, *Populus tremula* × *tremuloides* (all collected from Phytozome version12), *Arabidopsis* (collected from TAIR), and the putative *Q. suber* AMTs were aligned using ClustalW [46] using the following multiple alignment parameters: gap opening penalty 15, gap extension penalty 0.3, and delay divergent sequences set to 25%; the Gonnet series was selected as the protein weight matrix. Maximum-likelihood (ML) analyses were performed for the AMT family in a total of 74 protein sequences. Accession numbers of amino acid sequences are given in the Appendix A. Protein sequences were compiled and then aligned using the multiple sequence alignment tool MAFFT with an auto-model strategy [47] and then trimmed using trimAl v1.4.1 using the automated1 method [48]. An ML analysis was performed using MEGA-X v10.1.7 [49,50] following best model selection. The final tree was accessed using the LG substitution matrix model [51] with GAMMA-based thorough optimization. A final GAMMA-based thorough optimization of the best-scoring ML tree was performed. Bootstrapping was performed in the same program using 1000 replicates. Trees were visualized using FigTree version 1.3.1 [52].

### 4.5. Gene Expression Analysis by qPCR

The roots/leaves from two independent plants were pooled together, constituting one biological replicate. In total, three biological replicates from inoculated and non-inoculated plants were used for the qPCR analysis. Each biological replicate was grounded to a fine powder using liquid nitrogen. RNA was extracted using the method by [53]. Purified RNA samples were treated with DNase (TURBO DNA-free kit, invitrogen, ThermoFisher SCIENTIFIC, Cambridge, MA , USA) for genomic DNA removal, before cDNA synthesis with reverse transcriptase (RevertAid H Minus Reverse Transcriptase, ThermoFisher SCIENTIFIC) and oligo dT primer. qPCR was performed as described [54]. Briefly, primers for the specific amplification of cork oak transcripts involved in N transport and assimilation (AMTs, amino acid transporters, and GS/GOGAT pathway transcripts) were designed using PrimerSelect (DNASTAR). Primer specificity was tested using Primer-BLAST [55]. Diluted cDNA samples were amplified using SYBR Green (2X SensiFast SYBR Hi-ROX Mix, Bioline, London, UK) and the following thermal cycling conditions: initial denaturation at 95° for 10 min, followed by 40 cycles at 95 °C for 15 s and 55–64 °C (depending on the gene) for 30 s. Information on genes, primers, and qPCR amplification is shown in Appendix A. Melting curve analysis was used to identify non-specific PCR products (Appendix A). Target gene expression values were normalized to the expression of the elongation factor-1α gene (*EF-1α*) from *Q. suber* [54]. Quantification of transcript levels was calculated using the Schmittgen and Livak method [56]. qPCR amplicons were sequenced by Sanger sequencing. Significant differences among treatments were tested by Student’s t-test, using the IBM SPSS statistics 26 (Chicago, IL, USA).

### 4.6. Isolation of ECM Over-Expressed Q. suber AMT Genes and Functional Expression in Yeast

The full-length cDNAs from *QsAMT1.2-like*, *QsAMT3.2-like*, and *QsAMT3.3-like* (a) were obtained by PCR amplification of cDNA from ECM cork oak roots using the primers described in Appendix A and the proofreading enzyme Phusion High-Fidelity DNA polymerase (ThermoFisher SCIENTIFIC). PCR amplicons were subjected to agarose gel electrophoresis, and DNA bands were excised from the gel and purified (QIAquick Gel Extraction Kit, QIAGEN, Germantown, MD, USA). The purified DNAs were cloned into the pJET1.2/blunt Cloning Vector using the CloneJET PCR Cloning Kit (Thermo-Fisher SCIENTIFIC). Plasmids were then inserted into One Shot TOP10 chemically competent *E. coli* (ThermoFisher SCIENTIFIC). Following plating on LB agar medium supplemented with 100 µg/mL ampicillin, positive colonies were used for plasmid DNA purification by miniprep (GeneJET Plasmid Miniprep Kit, ThermoFisher SCIENTIFIC), and the constructs were verified by Sanger sequencing. The full-length *QsAMT1.2-like*, *QsAMT3.2-like*, and *QsAMT3.3-like* (a) cDNAs were subcloned into the yeast expression vector pDR196 using Gateway technology (Invitrogen, ThermoFisher SCIENTIFIC), as described earlier [28]. The resulting plasmids were called pDR196-*QsAMT1.2-like*, pDR196-*QsAMT3.2-like*, and pDR196-*QsAMT3.3-like* (a). The yeast strain 31019b (*MATa ura3 mep1*Δ *mep2*Δ::*LEU2 mep3*Δ::*KanMX2*) [29] was transformed with pDR196-*QsAMT1.2-like*, pDR196-*QsAMT3.2-like* or pDR196-*QsAMT3.3-like* (a) according to Dohmen et al. [57]. As a positive control, we also similarly cloned and transformed the low-affinity transporter *SbAMT3;1* from *Sorghum* as described by Koegel et al. [18]. The yest strain was also transformed with the empty pDR196 as a negative control. Transformants were selected on minimal YNB medium with amino acids as sole nitrogen source (drop-out mix lacking uracil) and with 2% glucose as carbon source. For the growth assays, yeast cells were grown on YNB without amino acids and with 2% glucose as the carbon source (pH 6.1). To this medium, nitrogen sources were added as required by the experiment. The nitrogen source used was (NH_4_)_2_SO_4_ at the specified concentrations. Complementation experiments were repeated 3 times for each tested AMT gene. Labeled N uptake studies were performed using [14C]-methylammonium (Amersham) (see Appendix A). Michaelis–Menten and Lineweaver–Burk representations of the data were used to determine apparent kinetics parameters (Km, Vm).

### 4.7. Subcellular Localization of Q. suber AMT Genes in N. benthamiana

The putative cork oak AMTs over-expressed in ECM root tissue were cloned using the Gateway^®^ technology (Invitrogen). To generate the *pDONR221-QsAMT1.2*, *pDONR221-QsAMT3.2*, and *pDONR221-QsAMT3.3* entry clones, the PCR fragments without the stop codon were amplified from cork oak cDNA using primers with the attB sequences (Appendix A). To accomplish the binary vectors, *pDONR221-QsAMT1.2*, *pDONR221-QsAMT3.2*, and *pDONR221-QsAMT3.3* were combined with *pGWB405* (Addgene) in a gateway LR reaction (Invitrogen) to generate *35S::*QsAMT1.2*::sGFP*, *35S::*QsAMT3.2*::sGFP*, and *35S::*QsAMT3.3*::sGFP* for subcellular localization. The GFP constructs were introduced in *Agrobacterium tumefaciens* (strain GV3101), and transient expression was performed by leaf infiltration [58]. *N. benthamiana* plants were grown in a growth room with long-day conditions at 25 ± 1 °C. Transiently transformed *N. benthamiana* were imaged two days after infiltration using a laser scanning confocal microscope (LSCS, Leica SPE). For each experiment, at least three independent infiltrations were performed. Two-channel image stacks were acquired sequentially on a Leica SPE confocal microscope using LAS X software, with a 63×, 1.15 NA objective. Pixel size in the XY obeyed the Nyquist sampling criterion. GFP detection was performed at 493–573 nm, and autofluorescence detection was performed at 573–745 nm; both channels excited using the 488 laser line.

### 4.8. Q. suber AMT Homology Modelling

To construct homology models for the AMTs upregulated in ECM roots (AMT1.2-like, AMT3.2-like, and AMT3.3-like (a)), the SWISS-MODEL web server [59] was used. The X-ray structure of EcAmtB (Pdb Id:1U7G) [31] was used as a template. The cork oak AMT sequences were first aligned to that of EcAmtB using ClustalW [46].

## Figures and Tables

**Figure 1 plants-12-00010-f001:**
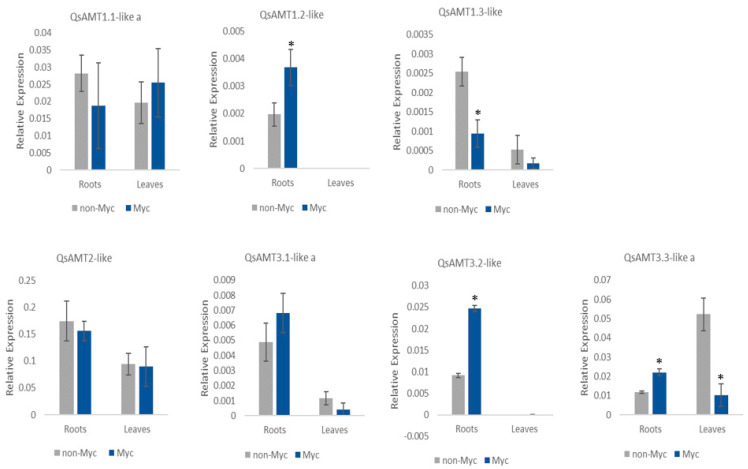
Quantification by qPCR of AMT gene expression in roots and leaves of *P. tinctorius* mycorrhizal (Myc) and non-mycorrhizal (non-Myc) cork oak plants. Values are the means of 3 replicates. Error bars represent standard deviations. The cork oak elongation factor-1-alpha transcript was used as reference. * Indicates significant differences between treatments at *p*-value < 0.05.

**Figure 2 plants-12-00010-f002:**
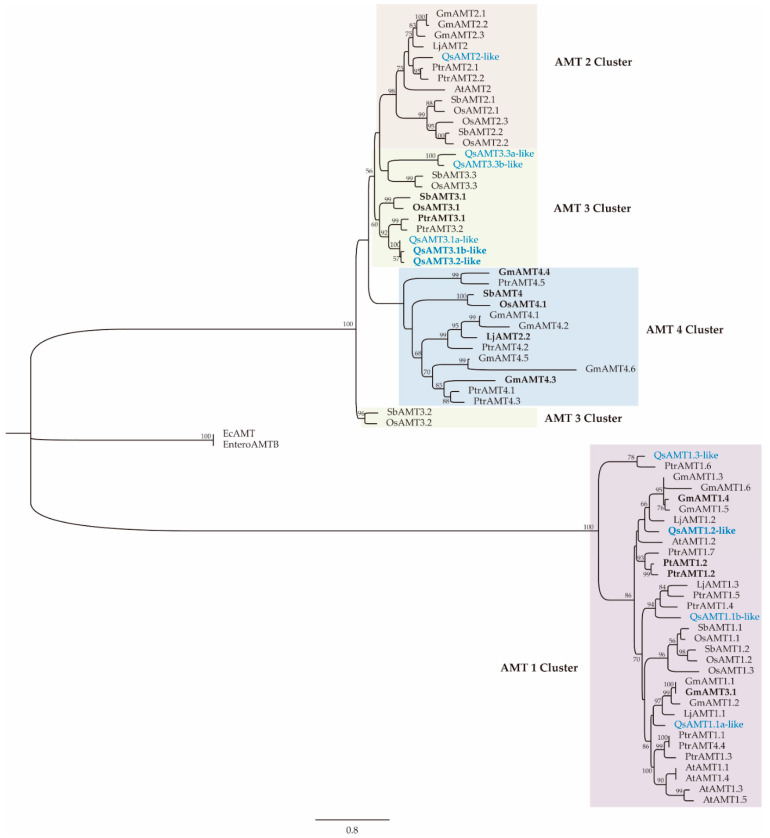
Maximum-likelihood tree of the ammonium transporter (AMT) protein family. Bootstrap was performed using 1000 replicated tests. Species codes: Ec, *Escherichia coli*; Sb, *Sorghum bicolor*; At, *Arabidopsis thaliana*; Gm, *Glycine max*; Lj, *Lotus japonicus*; Os, *Oryza sativa*; Ptr, *Populus trichocarpa*; Qs, *Quercus suber*; Pt, *Populus tremula*. Mycorrhiza-inducible AMTs are in bold font. *Quercus suber* AMTs are in blue font.

**Figure 3 plants-12-00010-f003:**
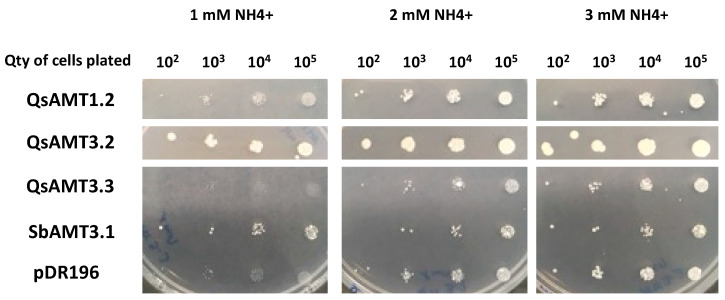
Complementation of a yeast mutant defective in ammonium uptake by QsAMT1.2, QsAMT3.2, and QsAMT3.3. This drop test shows the growth of the yeast strain 31019b, transformed with various constructs, on minimal medium supplemented with various NH_4_^+^ concentrations (1, 2 or 3 mM) as a sole nitrogen source. Quantity of yeast cells plated in each drop of 10 µL was between 10^2^–10^5^. All strains were incubated for 5 d at 29 °C. SbAMT3.1 from *Sorghum bicolor* was used as a control. pDR196 empty vector (control), *mep1*Δ*mep2*Δ*mep3*Δ + pDR196; QsAMT1.2, *mep1*Δ*mep2*Δ*mep3*Δ + pDR196-QsAMT1.2; QsAMT3.2, *mep1*Δ*mep2*Δ*mep3*Δ + pDR196-QsAMT3.2; QsAMT3.3, *mep1*Δ*mep2*Δ*mep3*Δ + pDR196-QsAMT3.3; SbAMT3.1, *mep1*Δ*mep2*Δ*mep3*Δ + pDR196-SbAMT3.1Δ.

**Figure 4 plants-12-00010-f004:**
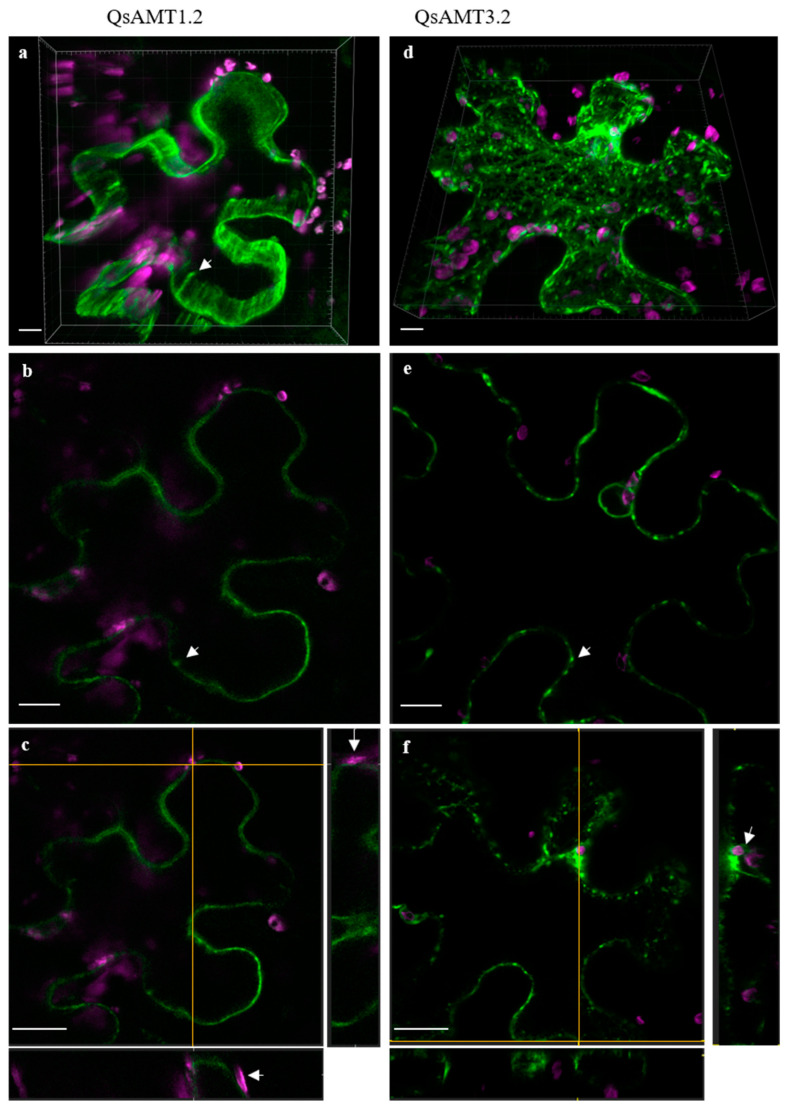
Cellular localization of QsAMT 1.2 and QsAMT3.2. (**a**) The 3D maximal intensity projection allows to visualize that the ammonium transporter 1.2 can only be localized in the plasma membrane with a stronger signal in structures that could reflect channels (arrow) (scale bar: 10 µm correspond to 1/5 of the larger grid square). (**b**) Same projection as in (**a**), where it is possible to clearly see that the QsAMT3.2 protein accumulates in specific spots in the plasma membrane, but it is also found in the cytoplasm, displaying the reticular pattern typical of protein transport. The signal is strong around the nucleus (scale bar: 10 µm correspond to 1/5 of the larger grid square). (**c**,**d**) Single confocal layer showing a stronger signal in the plasma membrane (arrows) of QsAMT1.2 (**c**) and QsAMT3.2 (**d**) (scale bar: 10 µm). (**e**) Orthogonal slide view of QsAMT1.2. Side and bottom images correspond to the vertical and horizontal lines crossing the central image, respectively. These side images are snapshots of what is being expressed in a precise point of the cell. In areas where chloroplasts emit autofluorescence (magenta), the signal is not detected in the cytoplasmic vicinity of the plastid, indicating that the tonoplast is not marked (arrow) (scale bar: 20 µm). (**f**) Same as in (**e**) but for QsAMT3.2, where plasma membrane spots can be perfectly distinguished. Surrounding the nucleus, we find some chloroplasts and cytoplasm marked by the reporter gene GFP (arrow) (Scale bar: 20 µm).

**Table 1 plants-12-00010-t001:** Amino acid residues forming the channel of the EcAmtB (template) and the mycorrhiza-induced QsAMT1.2-like, QsAMT3.2-like, and QsAMT3.3-like (a). Different residues with respect to the ones in EcAmtB are highlighted in red.

EcAmtB	QsAMT1.2-like	QsAMT3.2-like	QsAMT3.3-like (a)
Met23	Ala61	Gln40	Gln41
His168	His211	His205	His201
His318	His378	His359	His355
Leu114	Ile147	Leu151	Leu147
Leu208	Leu256	Leu245	Leu241
Ile266	Leu327	Ile303	Met299
Phe107	Phe140	Phe144	Phe140
Phe215	Phe263	Phe252	Phe248
Ile28	Phe66	Leu45	Leu46
Ser219	Ser217	Asp256	Asp252
Trp148	Trp181	Trp185	Trp181
Trp212	Trp260	Trp249	Trp245

## Data Availability

Not applicable.

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
