# Peer review of "Nitrogen Acquisition and Transport in the Ectomycorrhizal Symbiosis—Insights from the Interaction between an Oak Tree and *Pisolithus tinctorius"

_plants, 2022, doi:10.3390/plants12010010_

Round 1
Reviewer 1 Report
Excellent manuscript that shows highly relevant data, for the basic understanding of the phenomenon and also for applications, on the process of nitrogen acquisition and transport as part of the ectomycorrhizal symbiosis.
Author Response
We thank the reviewer for his/her kind words and support of our work.
Reviewer 2 Report
The Manuscript by M. Sebastiana et al. (plants-1993152) is dealing with nitrogen transport mechanisms in oak trees in interaction with an ectomycorrhizal fungus (Pisolithus tinctorius). The study reports expression analyses, taking advantage of the oak genome, of candidate genes from the plant that are potentially involved in nitrogen transport. Ammonium transporters have been found to be up-regulated in symbiotic interaction and subsequently functionally characterized using yeast complementation assays. The reported findings are interesting, complementing knowledge from other reported ectomycorrhizal associations regardin plant nutrient uptake.
Major points:
(1) Abstract: The Abstract should clarify from the beginning that expression of candidate genes was analyzed (lines 22-23).
(2) Even that the study and findings are interesting, it should be clearly discussed that the candidate gene strategy might miss other unexpected genes that could also be involved in symbiotic nitrogen transport.
(3) Induction of AMT gene expression has already been reported in other ectomycorrhizal models (see Introduction lines 85-87). So, this manuscript should more clearly state what is new here in this study and/or validation and extension of knowledge with another ectomycorrhizal model.
(4) Results: 2.1. lines 107-110, Even that these physiological parameters are presented within the Suppl. Table 1, it would be really interesting for the reader to understand more the context of this study if these short statements are explained somehow more in few details: meaning of the root colonization, age of plants / duration of experiment, experimental conditions regarding N concentration and number of plants...? Just some more (essential) information to understand these statements without going to the more detailed Methods section?
(5) Results: Within 2.3. (Phylogenetic analysis), one can read in line 145 "The ECM up-regulated cork oak AMTs" & in line 149 "The up-regulated QsAMT1.2-like", however, these "Results" were not yet described (this is reported later under 2.4.) and appear in this order really confusing. Please, check the logic order here for 2.3. & 2.4..
(6) Results: Within 2.4., tissue-specific gene expression with and without mycorrhiza, age of plants / duration of mycorrhization is missing. Lines 160-162, expression of AMTs, amino acid transporters, GS/GOGAT genes is announced, but Figure 2 presents (only) the AMTs, please precise. Moreover, in part 2.2. 10 AMT genes were stated for further analyses (line 118), but Figure 2presents (only) 8 genes? Please explain and/or correct. Mycorrhiza-effect and comparison of roots vs leaves should be better separated.
(7) Results: 2.5. & 2.6. Functional characterization in yeast and sub-cellular localization in epidermal cells deal with 3 or 2 AMTs, respectively. Mention at least the reason(s). But, with respect to these data, the former statement that 10 genes were further characterized should be revised and stated more precisely.
(8) Results: Figure 3 - Complementation appears only convincing in the case of QsAMT3.2. QsAMT1.2 seems to behave as the empty-vector control? The statement in line 200 (("1 mM and 2 mM.... (Figure 3)") should be revised, as Figure 3 shows only the 2 mM condition. Number of repetitions of this experiment is not given? To explain function or not function, it would have been interesting to test also the sub-cellular localiztion in yeast?
(9) Results: Regarding analyses of subcellular localization, size of Figure 4 should be increased for better resolution. Analyses and/or controls are needed to distinguish between plasma membrane and vacuolar membrane. From the presented pictures, the conclusions are hardly to follow.
(10) Results: 2.7. As it stands, comparison of the QsAMTs with the bacterial one sounds rather "Discussion" than "Result"? At least, the presented Table 1 would need illustration by a structural picture? The statement, that this sequence comparison would lead to "insights into the transport mechanisms" sounds very hypothetically and should be revised.
(11) What is your explanation/hypothesis concerning the reduction of QsAMT3.3 expression in leaves from mycorrhizal plants?
(12) Discussion: Transcriptional (up)regulation of certain transporters is indeed a strong argument for their involvement in a given function (here ammonium transport in mycorrhiza) as discussed by the Authors. However, non-(up)-regulation at the transcriptional level (eg lines 285ff and also within the final conclusion lines 353ff) will not reciprocally tell that such transporters are not involved as they might be regulated at different levels (translationally, functionally, localization....). Thus, some statements of the Discussion and part of the final conclusion should be revised.
Minor points:
(13) Introduction: The Introduction section can be clearly shorten avoiding repetitions of some statements (eg. regarding AM studies or nutrient benefits...). Some points should be reported more precisely, as "including on the molecular mechanisms underlying fungal N uptake" (lines 79-82). This statement needs to be corrected, as molecular transport mechanisms are even better characterized in ECM fungi compared to AM fungi (there are several references supporting this, eg. [11,16]). Moreover, the next phrase (lines 82-84) is clearly in contrast with this statement.
(14) Discussion: Overall, the Discussion section could be shorten (eg, the statement in lines 284-285 is repetition, paragraph lines 298ff repeats results, paragraph lines 310ff could be condensed...).
(15) Use of "ECM" is not properly defined, it is introduced in line 40 as "ectomycorrhiza", but further most often used as "ectomycorrhizal". Please correct.
(16) Meaning of (b) and (a) in lines 171, 172, 176,177 192, 200 is not clear.
(17) Figures: Figures 1 and 2 are hardly to read, these figures and also Figure 4 should be improved and size increased. If it's a matter of space, then maybe, Figure 1 (Phylogenetic tree) should go as Suppl. Fig.?
(18) Correct some spelling mistakes & spaces (eg. "cell" lines 73, 78; "S1" in line 470; phrase? line 209, "expressed" (line 215), correct also "transcripts" (line 215, expression of transcripts sounds wrong), phrase ? in lines 216-217 (we resorted to...?), "an" line 242, ...).
Reviewer 3 Report
Dear authors,
In this study oak plants were inoculated with the fungus Pisolithus tinctorius and used as a model system to investigate the molecular basis of N uptake and transfer between plant-fungus. Finding that ammonium is the main export form and characterizing some of the ammonium transporters involved.
The introduction is very well written, makes a good state of the art of this topic, and deals with the matter in due depth. The materials and methods are very well written and in sufficient detail.
Majors:
L26: “Functionality of these ammonium transporters” From the analysis of the results, I believe that this can only be said with certainty for one of them, QsAMT3,2. QsAMT3.3 and QsAMT1.2 do not complement the yeast mutant, and the Km of QsAMT1.2 is too high to be consider a specific ammonium transporter. And the sequence homology with the bacterial ammonium transporter is not high enough to be considered without a doubt a true ammonium transporter. Please rewrite this sentence
L114:” Twenty genes corresponding to AMTs were identified” Please indicate also in the results the basis on which this first selection has been made.
Figure 1 does not have enough definition to be read correctly, please improve it
L195: “QsAMT1.2-like and QsAMT3.2-like were able to efficiently restore the NH4+ uptake ability of the yeast mutant” Looking at the data in Figure 3, I do not see that QsAMT1.2 improves growth compared to the pDR196 control. I don't understand the reason for this statement. Please clarify
L197: “1 mM and 2 mM NH4+” In which figure are the data with 1 mM ammonia?
L200 :” In contrast, QsAMT3.3-like (a) was unable to clearly complement the NH4+ transport deficiency” please rewrite this sentence and comment that a negative effect is even observed with respect to the pDR1996 control, comment on its possible toxicity in yeast
L209:” revealed the to be, respectively,” I don't see sense, please rewrite it
L300: “which was sufficient to confer the ability to grow under low NH4+ concentration” Please rewrite this statement, based on the data in Fig 3, only QsAMT3.2 efficiently complements the yeast mutant, the other two do not.
Minors:
L70-L105: The length of this paragraph is too long, for a better understanding please, divide it into two or three paragraphs.
Figure 2: Y axis legend is missing
L170:” but at lower levels” except AMT3.3-like what is expressed more in the leaves, right? please rewrite
L176: “No induction of AMT transcripts was detected in leaves” please rewrite this sentence, I think you mean no induction by P. tinctorius was detected in leaves
Figure 3: In the figure legend indicated what is SbAMT3.1. Indicated that pDR196 is the control 31019b mutant. It is missing to indicate under the photo that 102, 103,.... are corresponds to the number of cells
L211: “667 µM (Figure S4).” Please replace by 667 µM respectively (Figure S4).”
L281” were unable to express QsAMT3.3 under our experimental conditions” please comment on possible toxicity
In the discussion the length of the paragraphs is also too long, please divide them into several
L437: (Figure S53)
L439: “1α gene (EF-1α) from Q. suber” indicate a reference of the reason for the election of this gene as normalizer
L470: see Methods S12
Reviewer 4 Report
This is a well-written manuscript that describe the mechanisms involved in nitrogen transport from ectomycorrhizal (ECM) fungi to their hosts, using the ECM fungi Pisolithus tinctorius colonizing Quercus suber host as a model. The methodology is appropriate and clearly demonstrates the role of ammonium transporters in this system. I suggest extending the discussion to put the results in the perspective of mycorrhizal communities with different N-uptake strategies. Otherwise I only have minor comments below:
Line 18
Not all trees associate with ECM fungi in temperate forests – please rephrase.
Line 20
What do you mean by event?
Line 24
Up-regulated when? After inoculation? Please rephrase.
Line 51
What about ECM fungi specialized in organic N uptake, such as Cortinarius?
Line 73
The abbreviations AM and AMT are confusing, I suggest modifying it.
Line 170
QS-AMT1.1 is not expressed at lower level in leaves and AMT1.3 expression is higher in non-inoculated plants. How do you explain these discrepancies? This would be interesting to discuss these results as well and not only focusing on the results that confirm the hypotheses that AMT are overexpressed in roots and inoculated plants.
Line 172
The use of “ECM plants” or “myc and non-myc” (also in the figures) is confusing since oaks are considered ECM plants in general. Do you mean inoculated and non-inoculated plants? My understanding is that RNA was extracted from the full root system of inoculated and non-inoculated plants (Line 422) or did you only take roots that show clear signs of colonization vs. roots that did not show signs of colonization? Do you expect the same results for gene expression in a non-inoculated plant and a non-colonized root segment of an inoculated plant?
Line 288
This work studied the effect of inoculation with one ECM fungal species that is known to be specialized in inorganic N uptake. Since there are several thousand ECM fungal species (some of them specializing in organic N uptake) I feel that the argument that “inorganic NH4+, and not organic N in the form of amino acids, could be the main N form delivered by ECM fungi to their host plants” is a bit strong. Do you think that different ECM fungal groups or ECM plant groups are specialized with a particular form of N transporter? What happens when a plant is colonized simultaneously by ECM fungal species with different N uptake strategies? Is N availability in the soil also inducing gene expression? Please develop.
Further readings include https://nph.onlinelibrary.wiley.com/doi/full/10.1111/nph.16322
Line 374
What do you mean by segments showing mycorrhizas? Presence of mantel and/or Hartig net?
Line 391
Please refer to the NCBI accessions. How did you make sure that the genes analyzed are single-copy and therefore truly orthologous?
Line 411
I suggest performing a maximum-likelihood tree search that takes into account a substitution model and is therefore more robust.
Line 422
What is the rationale for pooling samples from two independent plants into biological replicates instead of using 3 plants per treatment? How do you know that gene expression was similar before pooling them?
Round 2
Reviewer 3 Report
Dear authors,
The authors have accepted all my suggestions and I accept the current version of the manuscript as good.